# A bimodal soft electronic skin for tactile and touchless interaction in real time

Jin Ge [1], Xu Wang[1], Michael Drack[2], Oleksii Volkov[1], Mo Liang[1], Gilbert Santiago Cañón Bermúdez[1], Rico Illing[1], Changan Wang[1], Shengqiang Zhou [1], Jürgen Fassbender[1], Martin Kaltenbrunner [2,3] & Denys Makarov [1]

The emergence of smart electronics, human friendly robotics and supplemented or virtual reality demands electronic skins with both tactile and touchless perceptions for the manipulation of real and virtual objects. Here, we realize bifunctional electronic skins equipped with a compliant magnetic microelectromechanical system able to transduce both tactile—via mechanical pressure—and touchless—via magnetic fields—stimulations simultaneously. The magnetic microelectromechanical system separates electric signals from tactile and touchless interactions into two different regions, allowing the electronic skins to unambiguously distinguish the two modes in real time. Besides, its inherent magnetic specificity overcomes the interference from non-relevant objects and enables signal-programmable interactions. Ultimately, the magnetic microelectromechanical system enables complex interplay with physical objects enhanced with virtual content data in augmented reality, robotics, and medical applications.

[1] Helmholtz-Zentrum Dresden-Rossendorf e.V., Institute of Ion Beam Physics and Materials Research, Bautzner Landstrasse 400, 01328 Dresden, Germany. [2] Soft Materials Lab, Linz Institute of Technology, Johannes Kepler University Linz, Altenberger Strasse 69, 4040 Linz, Austria. [3] Soft Matter Physics, Johannes Kepler University Linz, Altenberger Strasse 69, 4040 Linz, Austria. Correspondence and requests for materials should be addressed to J.G. (email: j.ge@hzdr.de) or to M.K. (email: martin.kaltenbrunner@jku.at) or to D.M. (email: d.makarov@hzdr.de)

Electronic skins (e-skins) will revolutionize the way we interact with each other, with machines, electronics, and our surrounding environment[1–9]. Current systems here predominantly rely on interfacing either through physically touching (tactile interaction) or through tracking and monitoring of objects without approaching them (touchless interaction). The ever-increasing complexity that is involved in the manipulation of objects however calls for e-skins that are capable of simultaneously perceiving both tactile and touchless inputs[10–12]. Emerging technologies such as augmented reality (AR) appliances entail new requirements on the process of interaction that dissolve the now-common separation between tactile and touchless operation modes[11]. This paradigm shift will remove the barriers between handling virtual and physical objects and enable even complex interactions in a natural and intuitive way without the need for numerous regulation knobs and different sensory systems[10,13,14].

Assuring the best user experience when manipulating objects places several requirements on e-skins and their multimodal sensors. These forms of interactive electronics are ideally soft and mechanically compliant, selective in their response to objects of interest only and most importantly are able to unambiguously discriminate the desired interaction modes in real time. Presently, there are exciting demonstrations of compliant tactile[15–20] and touchless sensorics including humidity[21,22], magnetic[23–25], temperature[26], optical[27], and capacitive[28,29] sensors. Flexible capacitive structures[28,29] with both pressure and proximity detection are promising candidates for bimodal sensors. However, interference from irrelevant objects and difficulties in discriminating the signal source are fundamental challenges for capacitive systems. Combining individual tactile and touchless sensors on one flexible support is another appealing route. Yet, this requires significantly higher design efforts and often cumbersome and complex fabrication processes[27] that may hamper widespread permeation.

Here, we introduce a compliant magnetic microelectromechanical system (m-MEMS) enabling tactile and touchless interaction modes simultaneously in a single wearable sensor platform. The m-MEMS relies on a genuine, distinguishable bimodal sensing principle. It allows separating the signals from tactile and touchless interactions into two non-overlapping regions, realizing the challenging task of unambiguous discriminating the two interaction modes without knowing the history of the signal. The single sensing unit design of the m-MEMS not only simplifies sensor architecture for fabrication, but also avoids the interference from non-relevant objects. The magnetic touchless sensing mode of the m-MEMS is ready to specify the magnetic objects out of the irrelevant nonmagnetic objects and enables signal-programmable manipulation of the objects by adjusting the magnetic properties of objects of interest. Natural skin not only readily distinguishes different types of stimuli; it is also sensitive over a wide range of signal intensity. Implementations of tactile transducers are often optimized for a high-pressure sensitivity. Practical electronic embodiments would benefit from a high signal-to-noise ratio since only this readily allows for appropriate signal amplification and post processing. We thus optimized our m-MEMS to have a very high signal-to-noise ratio of above 100 in the pressure range from 0.72 to 11.6 kPa. Our m-MEMS e-skins enable complex interactions with a magnetically functionalized physical object that is supplemented with content data appearing in the virtual reality. We design and fabricate a demonstrator where our compliant m-MEMS skin is used not only to identify an object of interest but also to activate a pop-up menu and interact with its content relying on a combination of gestures and physical pressing. This intrinsically bimodal magnetosensitive smart skin allows reducing the number of physical "clicks" needed to activate the same functionality of the device to one, instead of at least three as up to now required when using state-of-the-art gadgets. The demonstrated enhanced—yet intuitive—interaction and manipulation ability enabled by our m-MEMS platform is an important milestone toward multifunctional, highly compliant human-machine interfaces. Beyond the field of AR, e-skins with multimodal interaction abilities are expected to bring benefits for healthcare, e.g., to ease surgery operations and manipulation of medical equipment[30,31], as well as for humanoid robots to overcome the challenging task of grasping[32,33].

## Results

**Compliant m-MEMS platform.** The m-MEMS platform is realized by packaging a flexible magnetic field sensor and a compliant permanent magnet with a pyramid-shaped extrusion at its top surface into a single architecture (Fig. 1a). The magnetic field sensor of the m-MEMS changes its electrical resistance when exposed to an external magnetic field of a magnetically functionalized object for touchless interaction and by mechanical deformation of the m-MEMS package upon application of pressure for tactile interaction (Fig. 1b). The signals of electrical resistance from tactile and touchless interactions separate into two non-overlapping regions by adjusting the field of the magnetic beacons in polarity and strength (Fig. 1c). The compliant m-MEMS platform consists of two major components. The first one is a soft frame based on a 335-μm-thick Polydimethylsiloxane (PDMS) rubber support with a central blind hole (Fig. 1d). The opening in the PDMS frame accommodates a 75-μm-thin compliant permanent magnet (NdFeB microparticles embedded in PDMS rubber) with 28-μm-high pyramid-shaped extrusions at its top surface (Fig. 1e–i). The second component is a high-performance magnetic field sensor, relying on the giant magnetoresistive (GMR) effect, which is hosted on a 20-μm-thin flexible polymeric foil (Fig. 1f). The thin foil seals the opening of the PDMS frame, resulting in a packaged flexible m-MEMS platform. Details on the fabrication are provided in Supplementary Figs. 1 and 2.

Applying a perpendicular directed pressure to the m-MEMS platform changes the distance between the permanent magnet and the GMR sensor through deformation of the 112-μm-thick air gap (Fig. 1g), which causes an altered field at the sensor location. The NdFeB microparticles inside the compliant permanent magnet (Fig. 1j and Supplementary Fig. 3) generate a magnetic stray field (Fig. 1k, Supplementary Figs. 4 and 5), which is adjusted to deliver a field strength in the range from about 2.1 to 1.7 mT at the location of the GMR sensor, depending on the magnet-to-sensor separation distance (Fig. 1l). This field range is selected to assure that the GMR sensor operates at its maximum sensitivity (Fig. 1m).

**Tactile and touchless operation modes.** The entire soft m-MEMS platform is thin and compliant (Fig. 2a), can be bent (Fig. 2b) and applied to curved surfaces such as a model finger (Fig. 2c). Worn as e-skin, our m-MEMS platform readily enables interaction with surrounding magnetic objects in both tactile and touchless modes (Fig. 2c). We demonstrate the capabilities with a delicate daisy flower to interact with (Supplementary Movie 1). One petal of the flower is decorated with a piece of thin and compliant permanent magnet (Fig. 2d) with a small field of <0.6 mT (Supplementary Fig. 6). The direction of the magnetic stray field generated by this patch is opposite to the polarity of the built-in magnetic field at the location of the GMR sensor that stems from the compliant permanent magnet of the m-MEMS platform. Figure 2e shows the time evolution of the change in

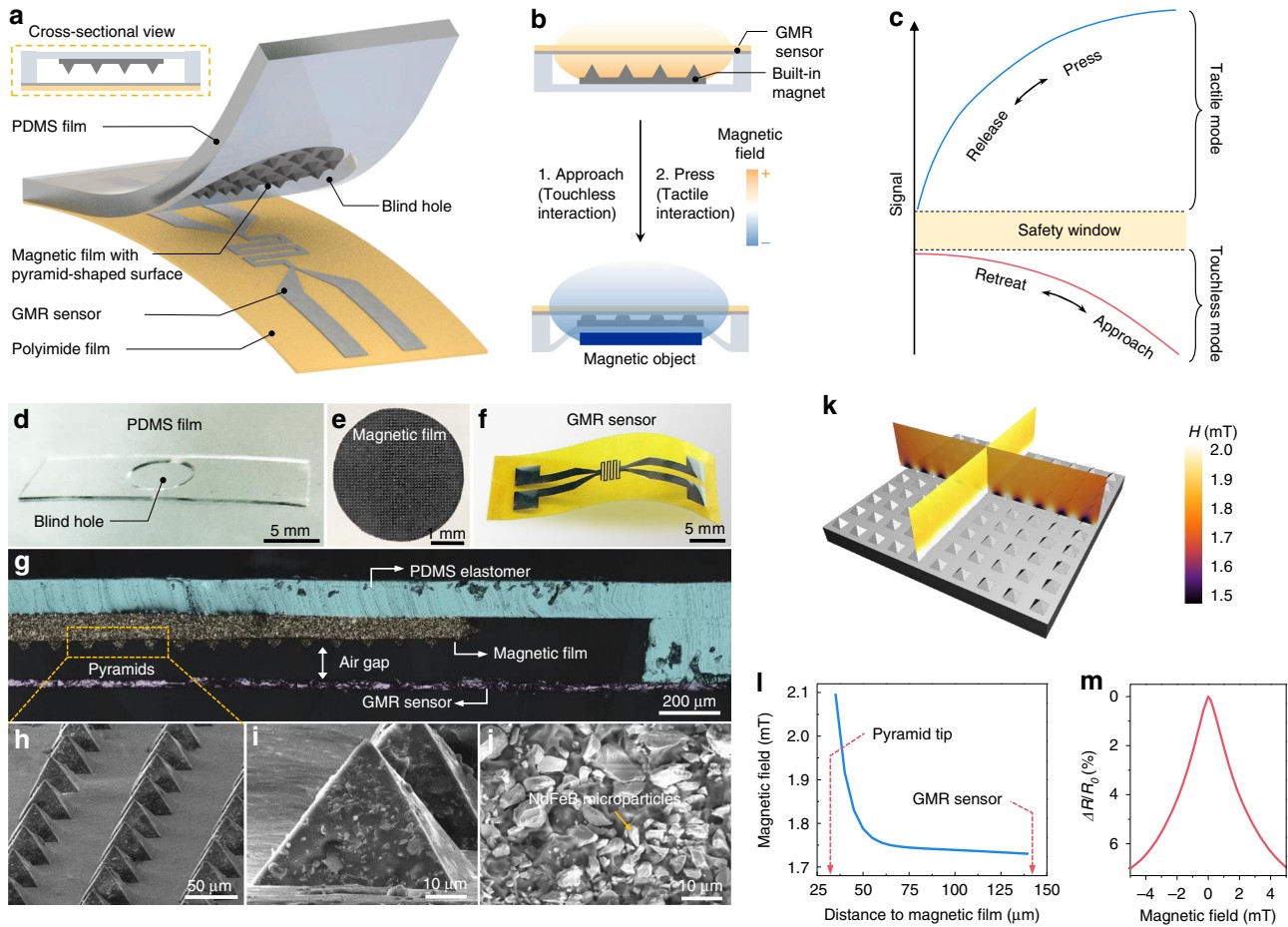

**Fig. 1** Assembly and sensing mechanisms of the compliant m-MEMS platform. **a** Schematic structure of the m-MEMS platform. **b** Mechanisms of the touchless (proximity) and tactile (pressure) sensing modes. The GMR sensor is biased by the built-in magnetic field (yellow cloud; only the field on the sensor side is sketched) of a compliant permanent magnet with pyramid-shaped extrusions. The m-MEMS is exposed to an external magnetic field with the opposite polarity (blue cloud) to the one of the built-in magnet. **c** The m-MEMS platform unambiguously discriminates touchless and tactile interaction modes, as the corresponding signals are located at different ranges with respect to a safety window (orange-shaded stripe). Since the signals are not overlapping, this method does not depend on the history of the interaction process (Supplementary Figs. 7 and 11). Optical images of (**d**) a PDMS frame with a blind hole, (**e**) a compliant permanent magnet with pyramid-shaped extrusions on its top surface, and (**f**) a GMR sensor on a polymeric foil. (**g**) Optical microscopy image (with false color) of the cross-section of the m-MEMS platform. SEM images of the pyramid-shaped extrusions in (**h**) low and (**i**) high magnification. **j** Cross-sectional SEM image of a compliant permanent magnet. **k** Simulated magnetic field profile above the compliant permanent magnet with pyramid-shaped extrusions (magnetized in a field of 1.5 T, Supplementary Fig. 5). **l** Simulated magnetic field profile between a pyramid tip and the GMR sensor. **m** The experimentally measured change of the electrical resistance ($\Delta R/R_0$) of the GMR sensor in response to an applied external magnetic field. $\Delta R = R_0 - R$, $R_0$ and $R$ are the initial and real-time resistance of the GMR, respectively

electrical resistance of the GMR sensor ($\Delta R/R_0$) upon approaching the petal, touching it and retracting the finger from the petal. The signal of the GMR sensor decreases when the fingertip approaches the petal (independent of the manipulation in the touchless mode, the signal remains negative; see discussion for the Supplementary Fig. 7). We note that the sensing responses might be different if the finger approaches the flower petal (Fig. 2a–e) from the back or from the front side. However, for in-plane isotropic magnetic stray fields varying with the distance to the object only, the readout will still be the same. As soon as the fingertip touches the petal and presses it, the signal rapidly increases and changes in sign to positive values where it remains during tactile interaction. Since the petal is very soft, the pressure applied on the petal is imperceptible by human skin and is estimated to be <1 kPa. Typically, the pressure used by humans for the manipulation of objects in the tactile mode is above 1 kPa (colloquially referred to as the softest human touch[34]). This qualitative signal change upon transition from touchless (negative

signal) to tactile (positive signal) interaction mode intrinsically renders discrimination between the modes unambiguous. Furthermore, in contrast to state-of-the-art reports, our m-MEMS skins sense the current interaction mode without the need to know the history of the signal change to interpret the state. This eases signal post processing and in turn speeds up response time.

The entire interaction process is illustrated in Fig. 2f. When the m-MEMS platform approaches the magnet on the petal, its field starts to compensate the built-in field of the compliant permanent magnet (Supplementary Fig. 7), resulting in the increase of the electrical resistance of the GMR sensor. The magnetic field will be compensated further until the m-MEMS platform is in touch with the magnetic object (the petal in this case). Applying pressure then moves the GMR sensor toward the pyramidal shaped built-in magnet. Here, the magnetic field at the location of the GMR sensor starts to increase, in turn leading to a decrease of the resistance of the GMR sensor. The processes illustrated in Fig. 2f are confirmed by the simulated change of the magnetic field at the

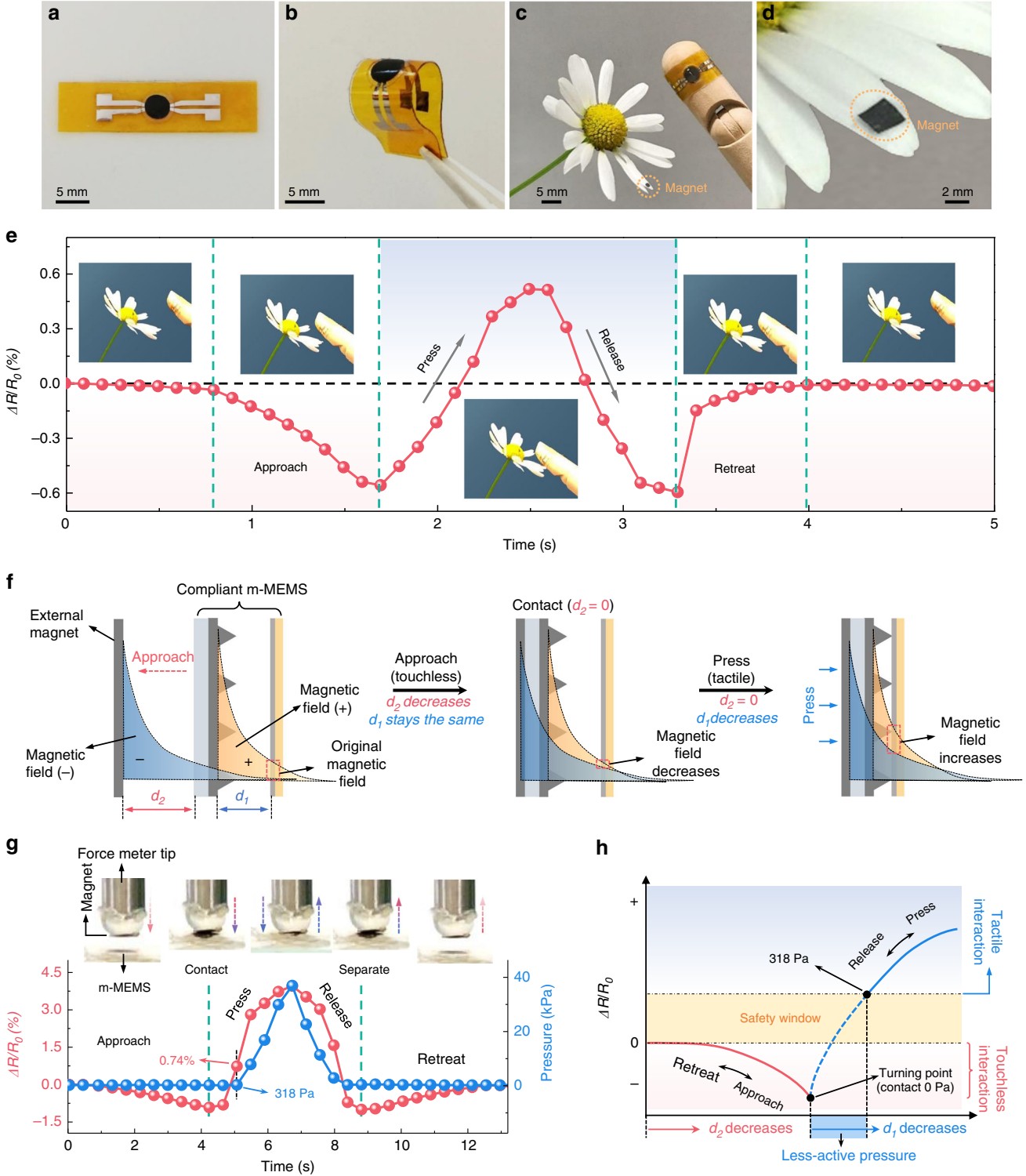

**Fig. 2** Tactile and touchless sensing with the compliant m-MEMS platform. Photographs of our m-MEMS platform in (**a**) a flat and (**b**) bent state. **c** A photograph of the MEMS platform wrapped around a wooden model finger. **d** A petal of a daisy flower decorated with a 20-µm-thin compliant permanent magnet (magnetized by 2.3 T, Supplementary Fig. 5). **e** Change of the electrical resistance ($\Delta R/R_0$) upon an interaction event where the finger bearing the m-MEMS platform approaches, touches and retreats from the magnet-decorated flower petal. The distance between the m-MEMS platform and the external permanent magnet is 4 mm. **f** Schematic illustration of the evolution of the strength of the magnetic field at the sensor location: superposition of the built-in magnetic field provided by the compliant permanent magnet (orange-shaded region) and an external magnetic field source (blue-shaded region). **g** Change of the electrical resistance (red dots) and the corresponding mechanical force (blue dots) applied to the m-MEMS during approaching, touching, pressing, releasing and retracting. **h** Schematic illustration of the safety window that separates the signal of the touchless interaction from the one of the tactile interaction

location of the GMR sensor during the interaction process (Supplementary Fig. 8).

We experimentally determine the applied pressure during tactile interaction and correlate it to the change of the electrical resistance of the GMR sensor ($\Delta R/R_0$) right after switching from touchless to tactile mode with a custom-built setup that simultaneously measures the change of the electrical resistance and the corresponding mechanical force applied on the m-MEMS platform (Supplementary Fig. 9 and Supplementary Movie 2). Here, the m-MEMS is moved upward to approach a permanent magnet (of same thickness and magnetization as the one on the flower petal, 4 mm in diameter) fixed at the tip of a force meter, followed by bringing them into contact and further pressing them together. When a predefined pressure is reached, we move the m-MEMS back to its original position. We evidence that the turning points in the resistance change are consistent with the ones of the force change (Fig. 2g). The small offset observed between the turning points in the force and relative resistance curves is due to the limited sensitivity of the force sensor (gentle pressure of <79 Pa at the beginning and end of the mechanical contact cannot be detected, Supplementary Fig. 10). Bringing the m-MEMS platform into contact with the magnet causes $\Delta R/R_0$ to jump from $-0.92\%$ (negative value, characteristic of touchless mode) to $+0.74\%$ (positive value, characteristic of the tactile mode) with a pressure increase of only 318 Pa, which is much smaller than the softest human touch (1 kPa).

The essence of manipulating magnetic objects with our m-MEMS platform is based on the algorithm that correlates the signal of the sensor $\Delta R/R_0$ to the commands for manipulation (Supplementary Fig. 11). As the signals specific to touchless and tactile manipulation modes do not overlap, the compliant m-MEMS is able to unambiguously discriminate between the two interaction types. In more detail (Fig. 2h), the value of $\Delta R/R_0$ upon touchless interaction is always negative, but $\Delta R/R_0$ at the initial stage of the tactile interaction (from the very event of touching to a certain degree of subsequent pressing) is also negative and thus in theory briefly overlaps in this region with the signal of the touchless interaction mode. However, in practical settings, especially such ones that involve manipulation processes with (human) fingers, the minimum pressure required to switch the signal polarity (here determined to be <318 Pa) is overcome immediately upon a pressing event. In addition, this transition to the positive signal range is achieved during <0.8 s when moving the finger even at a slow speed of 1 mm/s (Fig. 2g). We demonstrated that the evolution of the signal during the touchless and tactile interaction process is highly repeatable (Supplementary Fig. 12), which endows the m-MEMS platform with a safety window for reliably manipulating objects in both interaction modes.

**Enhancement of the m-MEMS performance**. The presence of the air gap between the GMR sensor and the compliant permanent magnet as well as the pyramid-shaped extrusions drastically enhance the performance of the compliant m-MEMS platform in terms of the pressure sensitivity and sensing speed. The pressure sensitivity is determined by the change of the magnetic field at the location of the GMR sensor, which is realized through relative displacement of the soft magnet in the blind hole (Fig. 1g). We simulate the mechanical deformation of the individual components (Fig. 3a) under low and high applied pressures. We extract the simulated change of air gap height versus pressure (Fig. 3b) from these calculations. Due to the cantilever-like structure of the m-MEMS package, the air gap height can be substantially reduced even at a low pressure of <764 Pa (Fig. 3a, upper panel), resulting in a sharp increase of $\Delta R/R_0$ for pressures over 318 Pa (see also

transition region in Fig. 2g, dashed blue line). At higher pressures (>764 Pa), the pyramid-shaped extrusions are brought into mechanical contact with the GMR sensor. Elastic deformation of the soft pyramids (Fig. 3a, lower panel) then brings the compliant permanent magnet still closer to the GMR sensor. This elastic shape change of the magnetic pyramids provokes a signal change $\Delta R/R_0$ in regions of higher pressures and significantly expands the sensitivity range of our m-MEMS platform (corresponds to the solid blue line in Fig. 2h) for object manipulation in the tactile mode. The dynamic shape deformation of the compliant m-MEMS platform under pressure is shown in Supplementary Movie 3.

To demonstrate the effectiveness of the air gap with pyramid-shaped soft magnet design, we compare the finite element mechanical simulations of the m-MEMS platforms possessing either only an air gap, or only pyramid-shaped extrusions (Fig. 3b). The displacement of the compliant permanent magnet towards the GMR sensor for the device with an air gap only (without pyramids) completes solely in the low-pressure range. In contrast, the displacement of the compliant permanent magnet towards the GMR sensor for the structure with pyramids only (no air gap) is very small in the low-pressure range. The simulated data are in agreement with the experimental results (Fig. 3c). The $\Delta R/R_0$ of the structure with air gap only and with both air gap and pyramids below 1 kPa is much higher than that of the structure with pyramids only, demonstrating the contribution of the air gap to the sharp increase of $\Delta R/R_0$ for pressures below the softest human touch. The change of $\Delta R/R_0$ for the structure with air gap only is higher than that of the structure with both air gap and pyramids for pressures below 1 kPa, but $\Delta R/R_0$ gradually saturates in the higher-pressure region rather than continuously increasing as in the case of structures with pyramids. This further corroborates the importance of pyramidal structures in improving the pressure sensitivity at high pressures.

Pyramidal extrusions in addition avoid sticking of the compliant permanent magnet to the PI foil of the GMR sensor. This anti-sticking effect significantly increases the dynamic range of the sensor, and ensures fast switching between touchless and tactile interaction modes in less than 75 μs (Fig. 3d, e). Without the pyramid-shaped extrusions, the compliant permanent magnet sticks to the surface of the PI foil and the electrical resistance of the m-MEMS recovers rather slowly after releasing the pressure (Supplementary Fig. 13).

Our m-MEMS platform exhibits an exceptionally high signal-to-noise ratio (SNR) of above 80 at the small pressure of 240 Pa, and above 100 in the entire pressure range from 0.72 to 11.6 kPa due to its small noise floor of only 0.01% (Fig. 3f). Our approach of optimizing the SNR rather than the pressure sensitivity is motivated by practical considerations of signal amplification. Indeed, from an electrical engineering point of view it is rather straightforward to amplify even small signals given that the SNR is large. We here outperform state-of-the-art reports[35–37] (Supplementary Fig. 14), rendering our m-MEMS concept highly promising even for pressure transducers only. With high SNR, our m-MEMS are able to detect different levels of pressure in a highly repeatable way (Fig. 3g). Even after 5000 pressure-release cycles (with a maximum pressure of 42.5 kPa), no degradation in the sensor response is observed (Fig. 3h).

**Array of compliant m-MEMS**. The compliant m-MEMS platform is readily scaled into a sensor array (Fig. 4a, b) for mapping the spatial distributions of magnetic fields and pressure. A PDMS frame is modified to have four openings (diameter of each opening, Ø, is 4 mm) accommodating four compliant permanent magnets (Ø = 3 mm, magnetized in an in-plane field of 1.5 T).

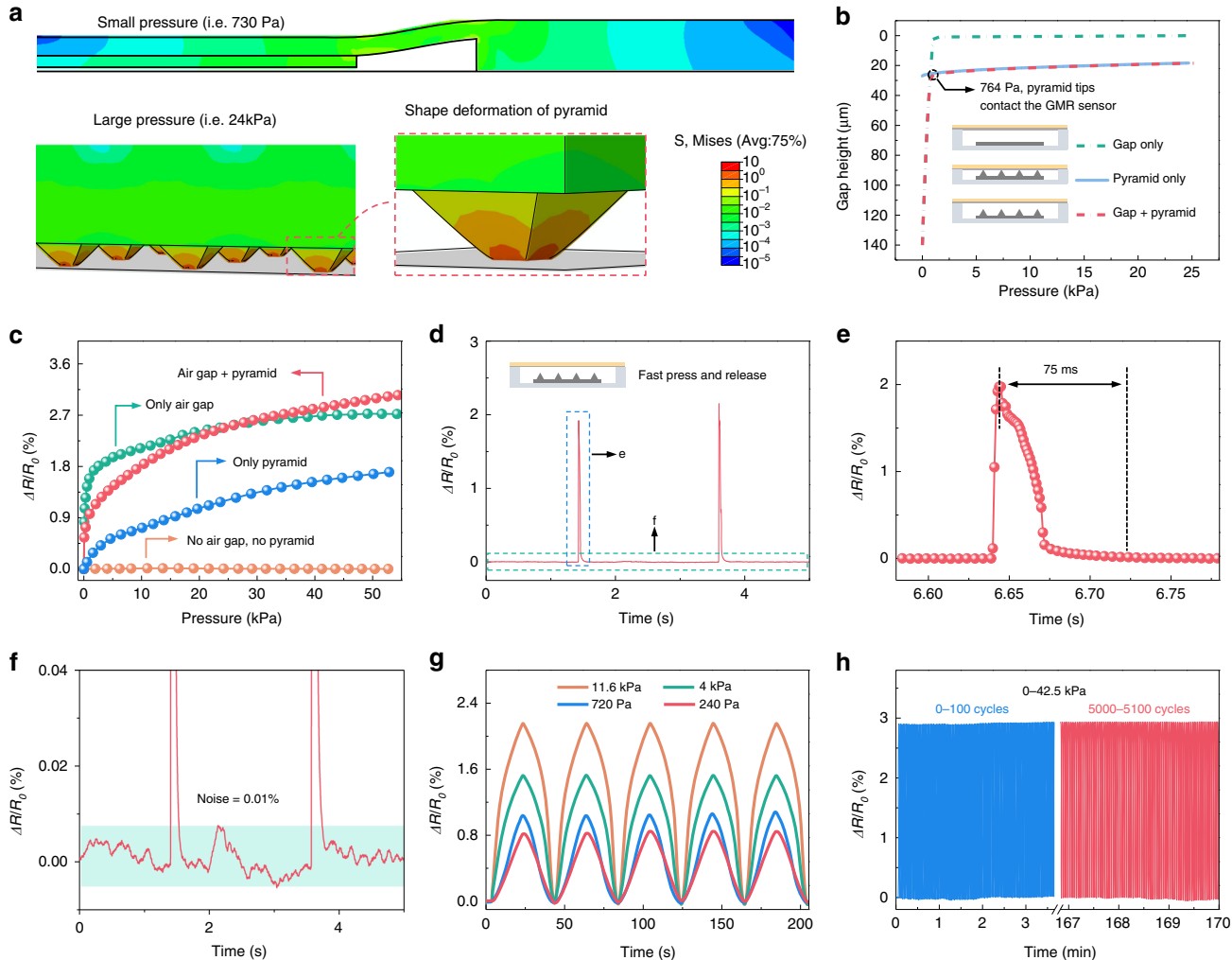

**Fig. 3** Impact of the m-MEMS structure on the pressure sensing performance. **a** Finite element mechanical simulations of the shape deformation and stress distribution of a compliant m-MEMS platform under high and low applied pressure. **b** The simulated change of the air gap height as a function of the applied pressure for the case of the m-MEMS platform possessing pyramids only, air gap only as well as a combination of air gap and pyramids. **c** The experimentally measured change of the electrical resistance of the compliant m-MEMS platform as well as the devices with pyramids only, with air gap only, and without air gap and pyramids, versus the applied pressure. **d** The experimentally measured change of the resistance of the compliant m-MEMS platform under a fast pressure stimulus. **e** The close-up of the area indicated with a dash square in **d**. **f** The close-up of the baseline in panel **d**, highlighting the excellent signal-to-noise ratio of over 100. **g** The experimentally measured change of the electrical resistance of the compliant m-MEMS platform during the press-release cycles. The device is exposed to a pressure of different magnitude of 240 Pa, 720 Pa, 4 kPa, and 11.6 kPa. **h** Durability test of the compliant m-MEMS platform by applying more than 5000 press-release cycles without signs of fatigue. Maximum pressure is 42.5 kPa

The PDMS frame is encapsulated with a polyimide foil hosting an array of four GMR sensors consisting of Py/Cu multilayers. Such packaged compliant m-MEMS arrays, here consisting of four functional elements (four-pixel array), intimately conform to curved surfaces such as a model finger (Fig. 4c). The four sensors are connected in series and driven with a constant current. The voltages of each m-MEMS are simultaneously recorded with a data acquisition box. The resistance of each pixel is calculated by dividing the measured voltage by the applied current. Based on the change of the resistance of each m-MEMS element upon applying a pressure or an external magnetic field, the distribution of both pressure and magnetic field can be uniquely recognized. We are thus able to map of the resistance changes of the m-MEMS in the array at every spatial position when activated by gently pressing (Fig. 4d). Here, a non-magnetic cotton swab is used to press one of the pixels in the array, leaving the others unaffected. Similarly, when an external magnetic object approaches each of the four m-MEMS pixels, the one closest to the

external magnet reveals the most pronounced change in resistance (Fig. 4e), thus enabling spatial mapping and contributing to the improved spatial resolution of the m-MEMS in touchless mode. Downscaling lateral dimensions of individual sensor elements can achieve further improvement of the spatial resolution.

**Multichoice 3D touch in AR**. AR allows to complement real objects with features and properties evident in virtual reality only. The possibility to interact with the virtual content and thus manipulate the properties of the real object is one of the major promises of future AR devices (Supplementary Fig. 15). We here realize an interactive e-skin with tactile and touchless sensing functionalities that allows us to perform complex interactions with a physical object (e.g., a region of a glass plate) supplemented with content data appearing in the virtual reality (e.g., virtual knobs superimposed on the glass plate). The data can be selected and manipulated using our compliant m-MEMS

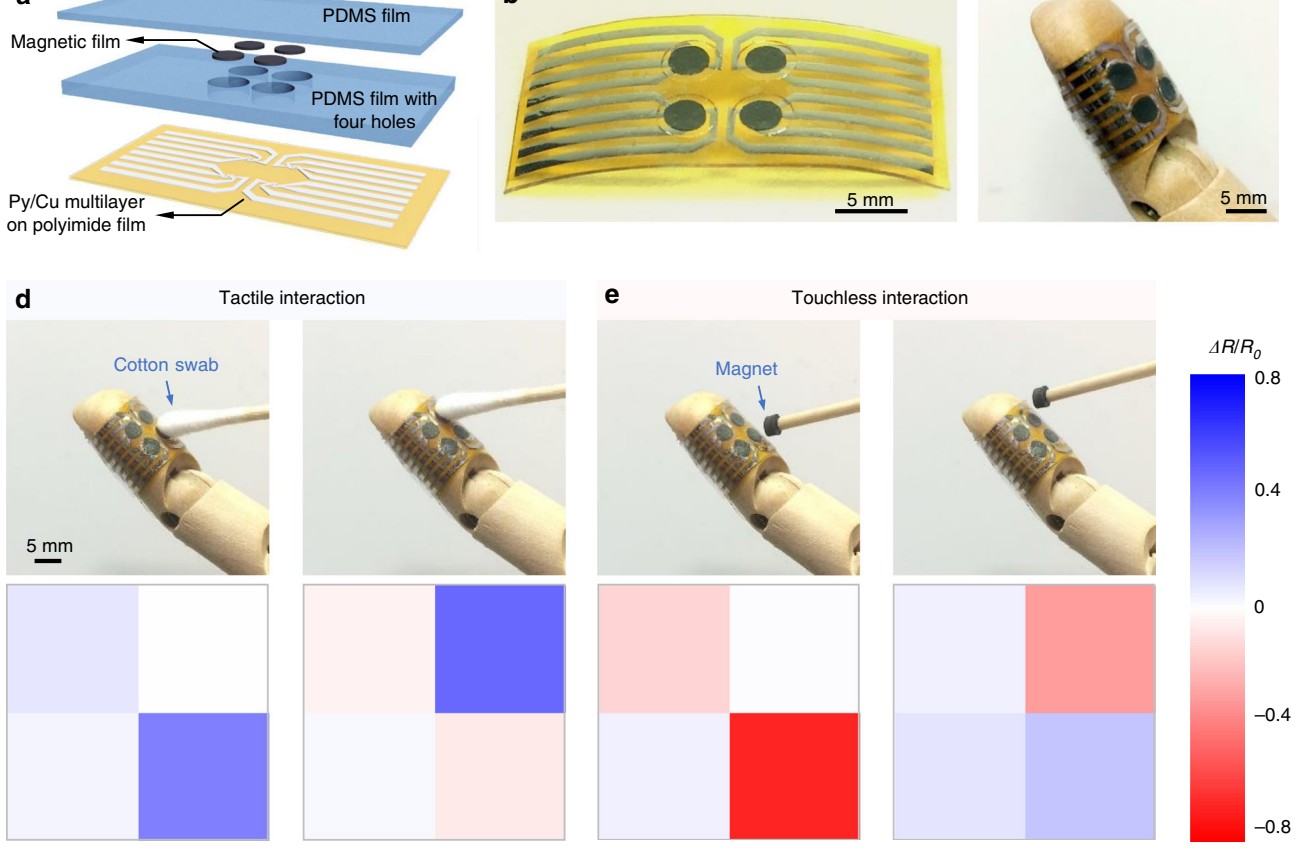

**Fig. 4** Array of compliant m-MEMS. **a** Schematic illustration of a 2 × 2 array of compliant m-MEMS. The PDMS frame is encapsulated with a polymeric foil hosting an array of four GMR sensors. **b** Photograph of a freestanding m-MEMS array. **c** A multi-pixel m-MEMS conformally adhered onto a model finger. **d** Resistance change $\Delta R/R_0$ of the four pixels when consecutively pressed with a cotton swab. **e** $\Delta R/R_0$ of the four pixels on the array when approached individually by a cylindrical soft magnet

platform in the form of multiple choices (Fig. 5, Supplementary Movie 4). Those pop-up at a (virtual) screen upon approaching the object, can be chosen by swiping the finger to the function of interest and can be manipulated by pressing. Finger-motion-correlated manipulation of multichoice in a touchless manner requires a sufficiently large interaction distance and thus sensitivity of the e-skin. Here, the real-world object (the glass plate) is equipped with a strong magnetic field source to improve the proximity sensitivity even at larger distances (Supplementary Figs. 16 and 17).

To demonstrate the concept, we adhere a compliant m-MEMS platform conformally to a wooden fingertip (inset in Fig. 5f) that then interacts with a glass plate with a permanent magnet located behind. Since the distance of the touchless interaction increases with a decreasing gradient of the magnetic field in the front region of the glass plate, we fix a permanent magnet with a strong magnetic stray field 1-cm-far behind the glass plate in order to have a long touchless interaction distance (Supplementary Fig. 18). We here arrange four GMR sensors into a Wheatstone bridge and package one of them into the m-MEMS platform to cancel unwanted thermoresistive effects of the GMR sensors (Supplementary Fig. 19 and Supplementary Note 1). Although this translates the output signal into voltage changes ($\Delta V/V_0$) rather than a change of resistance ($\Delta R/R_0$), the signal $\Delta V/V_0$ generated through touchless manipulation and tactile manipulation is still separated by a safety window and can be unambiguously discriminated (Supplementary Fig. 20). Upon approaching the glass plate, the $\Delta V/V_0$ of the m-MEMS platform increases. Once the m-MEMS sensor contacts the glass plate and pressure is

applied, the $\Delta V/V_0$ starts to decrease. Our m-MEMS e-skin is able to select a virtual item (i.e., room temperature, Fig. 5b–d, g–i, l–n) from a multichoice list on a virtual display in touchless mode, and then adjust the value of this option using tactile interactions between the finger and the glass plate (Fig. 5e, j, o). For example, the room temperature can be decreased by firmly pressing or increased by gently pressing.

The manipulation of virtual objects is realized by coding different execution sequences triggered by electrically read signals ($\Delta V/V_0$) from the m-MEMS. Figure 5p shows the evolution of the electrical signal during the interaction process between a wooden finger and a glass plate decorated with a piece of permanent magnet (Fig. 5a–o). We define distinct voltage thresholds and motion patterns of the finger and correlate them with switching events of data elements shown on a virtual screen. The option of interest (here, room temperature) is selected for further manipulation by pointing to a proper location for a while. When $\Delta V/V_0$ passes the safety window, the tactile interaction mode is activated and $\Delta V/V_0$ is used to adjust the value of the visual item of choice. For example, by firmly pressing, the temperature is changed from 21.0 °C to 19.6 °C. Our e-skin here enables us to manipulate virtual objects in a way we usually do with physical objects.

The field of a magnetic object can be adjusted to the requirement in terms of its strength and gradient (Figs. 2 and 5). Thus, the electrical signal of the touchless interaction can be defined for easily coding the manipulation of the objects of choice, which enables intrinsic selectivity upon interaction with irrelevant objects. Our multichoice 3D touch demonstrator (Fig. 5, Supplementary Movie 4), may enable a broad range of applications. Beyond

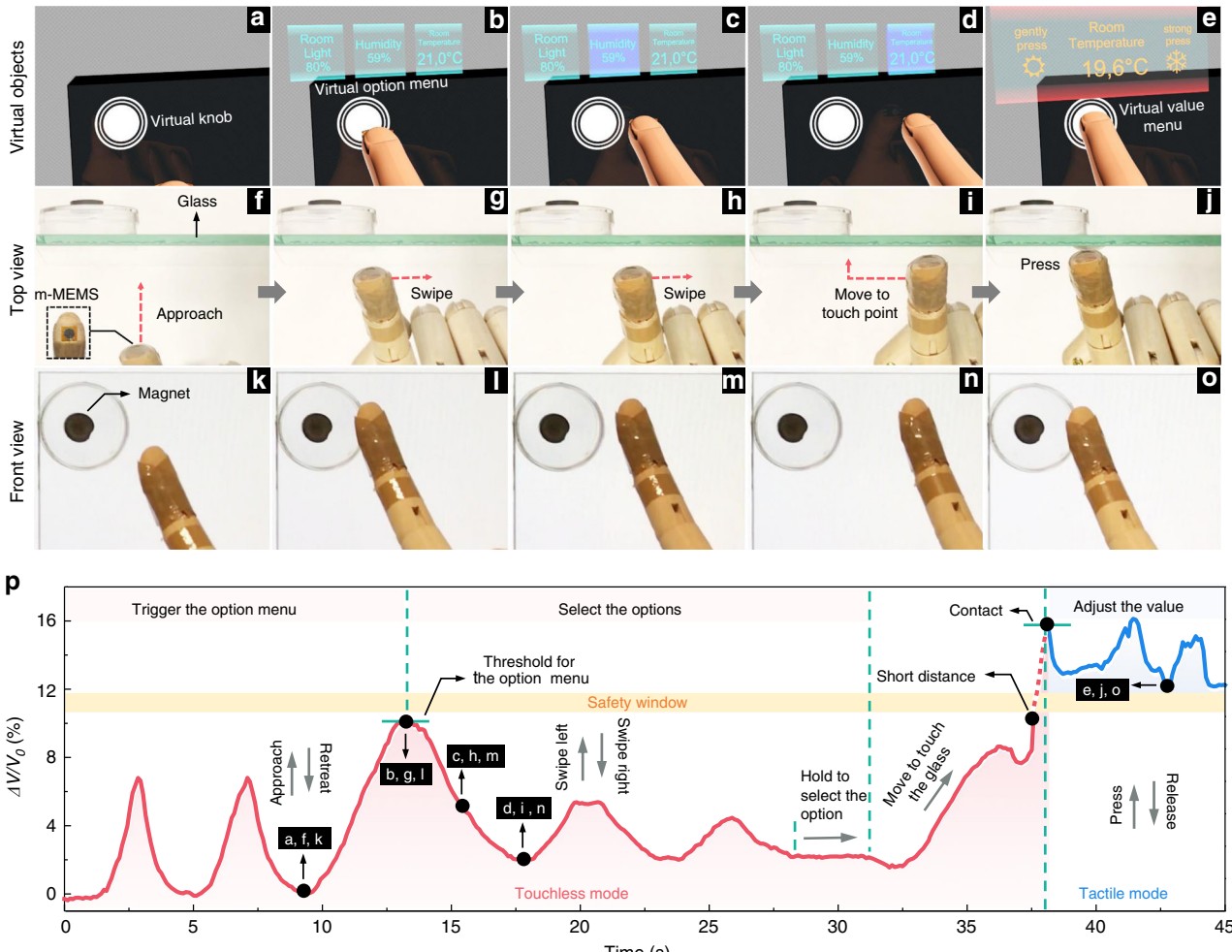

**Fig. 5** Multichoice 3D touch in augmented reality. **a–e** The interactive on-skin device with tactile and touchless sensing functionalities allows us to perform complex interactions with a physical object (e.g., a functionalized region on a glass plate) supplemented with a content data appearing in the virtual reality (a virtual knob). The actual realization of the concept is shown with snapshots of the Supplementary Movie 4 shown in **f–j** top view and **k–o** front view where an m-MEMS applied to a fingertip approaches the virtual knob, swipes over it and proceeds with pressing. As the first step (**a, f, k**), the e-skin interacts with the knob in a touchless way upon approach. At a certain distance to the object, a selection of options appears on a virtual display (**b, g, l**). The options are pre-assigned to this object, e.g., room light on/off, humidity, room temperature. The corresponding option (e.g., temperature) can be chosen by a gesture, e.g. motion of a finger towards the object of choice (**c, h, m** and **d, i, n**). Its value can be further manipulated using conventional tactile interaction by pressing the knob with a finger (**e, j, o**). **p** Evolution of the electrical readout upon the interaction process between the compliant m-MEMS platform applied to a wood finger and a glass plate, decorated with a permanent magnet. $\Delta V = V_0 - V$ with $V_0$ and $V$ denoting the initial voltage and real-time voltage, respectively

proximity sensing, our m-MEMS platform also features an angle sensing functionality (Supplementary Fig. 21, Supplementary Note 2, Supplementary Movie 5), which is of advantage for rotation-based manipulations in AR settings[25,38] and medical applications. Generally, we envision that surgeons wearing e-skins equipped with an m-MEMS platform will acquire the ability to quantitatively sense mechanical properties of tissues upon a regular palpation. At the same time, the touchless interaction functionality may enable surgeons to manipulate medical equipment in a touchless manner, helping them to avoid unwanted contamination. Furthermore, touchless perception through our m-MEMS platform may enable robots to navigate their grippers to a desired position in a touchless way, while the tactile perception guarantees a suitable force for grasping objects without dropping or breaking them. Although the demonstrations here are done with the m-MEMS platform positioned on a fingertip of a wooden finger, the same performance is achieved when the device is positioned on a human finger (Supplementary Fig. 22, Supplementary Movie 6).

## Discussion

We put forth a concept of e-skins equipped with a compliant magnetic microelectromechanical system (m-MEMS) that synergistically combines tactile and touchless interaction modes in a single sensor unit. The unambiguous discrimination between the interaction modes is aided by the rational design of the structure (air gap and pyramid-shaped extrusions) and the magnetic stray field inside the m-MEMS as well as the one around an object of interest. The m-MEMS architecture in this work is of a general design. Albeit already highly functional, further performance optimization is expected through adjusting size and number of the pyramids, the gap height, the thickness of the PDMS film, the diameter of the circular opening and the magnetic moment of the compliant magnet.

We showcase the usability of our bimodal e-skin in AR settings, where a sensor-functionalized hand performs complex selection and manipulation of virtual objects by simultaneously using the two sensing modes. Our concept provides a fertile base

for a cornucopia of applications in interactive electronics, supplemented reality, human-machine interfaces, but also for the realization of smart soft robotics with highly compliant integrated feedback systems as well as in medicine for physicians and surgeons.

Our here-developed m-MEMS platform is very general and can be realized based on magnetic field sensors other than those relying on the GMR effect. Concepts including compliant Hall effect[24,39] (sensitive to out-of-plane magnetic fields) and compliant planar Hall effect[40] (sensitive to in-plane magnetic fields) sensors might offer further advantages to m-MEMS platforms. They are linear sensors of magnetic field and also possess high sensitivity to small magnetic fields. Considering that generic stray fields of magnetized objects contain three components (two in-plane and one out-of-plane), we envision that the m-MEMS would benefit from three-axial compliant magnetic field sensors.

The wearable m-MEMS platform supports interaction with multiple magnetic objects. If the functionality of objects is encoded in a different magnetic field profile (different field strength, different symmetry of the magnetic field, different field gradients), then our sensing platform can be trained to recognize these objects. This is the case demonstrated in Figs. 2 and 5. In the first case when a thin magnetic patch is used, the field is weak and interaction starts at close distance. In the second case (standard permanent magnet), the field is rather strong and interaction starts at a larger distance to the object. The spatio-temporal variation of the signal will be different when approaching these two objects. These differences are unique fingerprints of each magnetic object, allowing constructing spatio-temporal maps for their unique identification.

## Methods

**Preparation of the silicon mold**. Thermally oxidized $SiO_2$ (1000 nm)/Si(100) wafers were coated with a grid photoresist pattern relying on a regular photolithography processing. The exposed $SiO_2$ patterns were etched by HF solutions. After this, the samples were anisotropically etched in the solution of KOH and isopropanol (35%wt KOH in $H_2O$: Isopropanol = 4:1, v/v) at 80°C. When the etching process was finished, the samples were cleaned in water and ethanol, and further modified with 1 H,1 H,2 H,2H-perfluorodecyltrichlorosilane (Sigma-Aldrich) by gas phase silanization to prevent adhesion.

**Preparation of the GMR sensor based on Py/Cu multilayers**. Polyimide (PI) resin (PI2545, MicroSystem, USA) was drop casted on Polyethylene terephthalate (PET) sheet (125 μm thick) fixed to a film applicator (TOC AB3400). Then, a wet PI coating was fabricated by the film applicator. The wet polymeric film was dried at 80 °C. After drying, the sample was heated at 200 °C for 1 h to crosslink the PI film. Here, PI films with a thickness of 20 μm were used as a substrate for the deposition of GMR sensors. GMR multilayers with a stack [substrate//Ta(5 nm)/[Py(1.5 nm)/Cu(2.3 nm)]$_{30}$/Py(1.5 nm)] were grown by magnetron sputter deposition (BESTEC, Germany) at room temperature.

**Characterizations**. Scanning electron microscopy (SEM) images and elemental mappings were taken using a Hitachi S-4800 microscope. Magnetic hysteresis loops were measured at 300 K using a superconducting quantum interference device vibrating sample magnetometer (SQUID-VSM, Quantum Design). Compliant permanent magnets (NdFeB microparticles embedded in PDMS) were magnetized in a 1.5 T magnetic field of an electromagnet. Confocal microscopy images were taken using a confocal microscope (Zeiss, Smartproof 5). Giant magnetoresistive performance of Py/Cu multilayers (Py: $Ni_{80}Fe_{20}$) was measured at room temperature in an in-plane magnetic field generated by an electromagnet. The force was measured by a universal digital force gauge (Sauter FH-5). Electrical resistance of the compliant m-MEMS platform was measured using a Keysight B2902A or Keysight 34461 A device.

**Magnetic field simulation**. Here, we describe our calculation of the spatial distribution of a magnetic field $\vec{B}$ outside a disk-shaped compliant permanent magnet with pyramid-shaped extrusions, which is homogeneously filled with magnetic NdFeB microparticles. To simplify numerical calculations, we assume that each microparticle is (i) of spherical shape with a radius, $R$, of 2.5 μm and (ii) magnetized along $\vec{y}$-axis. In this case, the magnetic field, which is generated outside each

magnetic particle, has a following form:

$$\vec{B}_i(\vec{r}_i) = \frac{\mu_0}{4\pi} \frac{3\vec{n}_i\left(\vec{n}_i\vec{M}\right) - \vec{M}}{r_i^3}$$

where $\mu_0$ is a vacuum permeability, $\vec{n}_i = \vec{r}_i/r_i$ with $r_i = \sqrt{(x-x_i)^2 + (y-y_i)^2 + (z-z_i)^2}$ being a distance from the center of a magnetic particle $(x_i,y_i,z_i)$ to any other point $(x, y, z)$, $\vec{M} = \frac{4}{3}\pi R^3 M_s \vec{y}$ is the total magnetic moment of a spherical particle, with $M_s$ being the saturation magnetization.

Due to the big density of magnetic microparticles inside the polymeric disk-shaped film and their random distribution, we consider the magnetic disk as a homogeneously magnetized body with a reduced saturation magnetization compared with an individual microparticle. We calculate numerically the spatial distribution of the magnetic field generated by a disk and sum it with a field distribution, which is generated by magnetic microparticles located in pyramids. The resulting field distribution is presented in Fig. 1k.

**Mechanical simulations**. The mechanical simulation has been calculated in Abaqus Standard. It is split into the deformation of the sensor enclosure with rotational symmetry and the deformation of the NdFeB permanent magnet pyramids. The sensor geometry as shown in Supplementary Fig. 9 is indented in perpendicular direction by a rigid stamp of 4-mm diameter that is loaded by a certain pressure. The two cases of a non-sealed inner sensor volume with unhampered air exchange as well as a perfectly sealed volume with adiabatic compression of the air content have been simulated. For the compression of the permanent magnet, a single unit cell of an infinite array of the pyramid-structured disk has been simulated. For both cases the displacement versus the applied pressure was calculated.

**Discrimination of the two interaction modes**. Upon approaching or retreating the finger with a m-MEMS to or from an external magnetic object (blue object), the electrical resistance of the magnetic field sensor integrated in the m-MEMS platform will change (Supplementary Fig. 7). In the touchless mode, the signal change remains negative (Fig. 2 and Supplementary Fig. 11). The software decides that the interaction mode is switched from the touchless to the tactile one as soon as the signal $\Delta R/R_0$ reached the contact point (0 Pa; highest negative signal change) and subsequently raised up to the value corresponding to 318 Pa of applied pressure. As soon as these two conditions are detected by the electronics, the signal analysis is paused for a certain user-defined time. After this waiting time, the signal level is analyzed further. If $\Delta R/R_0$ is positive (negative), the interaction mode is assigned to be tactile (touchless) (Supplementary Fig. 11).

We note that the compliant m-MEMS platform can provide a reliable and unambiguous separation between the two interaction modes even if the settings are sub-optimal. One of the examples is when the strength of the external magnetic field is chosen to be large enough that the resulting resistance change upon touchless interaction is very large and positive values of the resistance change cannot be achieved upon tactile interaction. Still, by properly choosing the logic behind the interaction process, touchless and tactile modes can be readily separated (Fig. 5 and Supplementary Fig. 20).

## Data availability

The datasets generated during and/or analysed during the current study are available from the corresponding author on reasonable request.

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

## Acknowledgements

We acknowledge T. Voitsekhivska, R. Kaltofen, Dr I. Mönch, B. Scheumann, G. Schnabel (Helmholtz-Zentrum Dresden-Rossendorf e.V.) for their assistance in the sample preparation; Dr T. Kosub (HZDR) for the support with the magnetotransport characterization; E. Christalle and Dr R. Hübner (HZDR) for the SEM and EDX measurements; Z. Zhang for the assistance in preparing the software for the data acquisition. Support by the Structural Characterization Facilities Rossendorf at the Ion Beam Center (IBC) at the HZDR is greatly appreciated. This work was financed in part via the German Research Foundation (DFG) grants MA 5144/9-1 and MA 5144/13-1, the European Research Council Starting Grant "GEL-SYS" (grant agreement no. 757931) and a startup grant of the Linz Institute of Technology (grant agreement no. LIT013144001SEL).

## Author contributions

J.G. conceived the idea, designed, and carried out the experiments. J.G., M.K., and D.M. analyzed the results and wrote the paper with contributions from all authors. O.V. performed magnetic simulations. M.D. performed mechanical simulations. X.W. and M.L. performed the optimization of compliant permanent magnets. J.G. and G.S.C.B prepared the software and hardware for the data acquisition. J.G., R.I., and D.M. designed multichoice 3D touch demonstrator. C.A.W. and S.Q.Z. measured the integral magnetic properties of the samples. All co-authors discussed the results and edited the manuscript. D.M., M.K. and J.F. supervised the project.

## Additional information

**Competing interests:** The authors declare no competing interests.

