## [Peer Review File · Nature Communications]

Reviewers' comments:

Reviewer #1 (Remarks to the Author):

In this paper, Ge et. al. present a bimodal electronic skin that can detect tactile and touchless interactions with an external magnet. The sensor composed of a GMR sensor and a soft composite magnet with micro pyramid arrays. The manuscript is nicely written, and the figures are beautiful. The idea of combining tactile and touchless interactions using all magnetic sensors is interesting, which could expand the sensing regime of conventional electronic skins. I also like the optimization of the pyramid shape soft magnet to optimize the sensitivity for the GMR sensor. However, the system has a few major challenges for the exciting future applications authors visioned in the paper.

1, Limited sensing pose in the touchless sensing mode.

In the demonstrations (figure 2), the magnetic fields generated by the magnetic film and magnetic object are always the opposite. This requires a fixed angle between the sensor and the external magnet. However, in the applications of surgical operations and robotic grasping (as mentioned in the introduction), the orientation sensing of the target object is crucial for manipulation. How can this system sense the orientation? For example, in figure 2, what will happen if the finger approaches the flower paddle from the back or from the side? Can the sensor still separate the tactile and touchless modes?

Without the capability of sensing the orientation, the system simply just using a well-studied magnetic force sensor (embed magnetic in a soft matrix with a magnetic field sensor) to sense an external magnet.

2, Number of magnetic objects this sensor can distinguish.

In programmable interactions (mentioned in the abstract), the electronic skin needs to interact with multiple target objects. I wonder how the current e-skin distinguish different magnetic objects. Is that possible to separate magnetic objects by strength, orientation, or combinations of multiple magnets?

In the manipulation of virtual objects, as in Figure 5a, the finger can select virtual option menu in the touchless mode by swipe the finger. I assume the sensor chooses the option by the magnetic field strength. If that is true, moving away the sensor will have the same effect, making it unfit to manipulate virtual objects due to limited spatial resolution.

The author can provide a systematic discussion on the spatial resolution (position and orientation) of this sensor in the touchless mode, and comment on how to solve this issue.

3, Sensing on a soft substrate and spatial of the sensors

Considering the size of the sensor (~5mm), the surface under the sensor region cannot be simply considered rigid. In the demonstrations (figure2,4,5), the sensors are wrapped around a rigid wooden finger. I am just wondering what will happen on a human finger, and how much the soft substrate can impact tactile sensing signals. The spatial sensing resolution at the x-y plane (parallel to the skin) is significantly smaller than the z direction (perpendicular to the skin). Can miniaturization solve this issue? Maybe the authors can comment on this as a perspective.

Here are also some minor comments:

- 1, Some expressions are quite dramatic, e.g. "Practical electronic embodiments would benefit most ..." "... our m-MEMS platform is a key milestone ..."
- 2, I don't understand "magnetic functionality", I suggest changing it to "embedded magnet".

Reviewer #2 (Remarks to the Author):

This manuscript describes the use of a GMR (giant magneto resistive) sensor, which changes resistance in response to magnetic fields. Here, the magnetic fields come from two sources. One

source is an external magnet, which can be detected in proximity without direct contact. The second is from an array of magnetic (soft) pyramids that hover over the sensor. When this array is pressed, the pyramids are brought into closer proximity to the sensor and can therefore be used for sensing touch. This type of bi-modal sensing is of interest for prosthetics and human-machine interactions, for example.

Although the paper is well-written with nice figures and demonstrations, my concern is the following. The use of GMR for sensing remote magnetic fields has been done previously. The use of pyramids for sensing touch has also been done previously, but using capacitance. If I understand, the novelty is (1) the use of pyramids for GMR sensing touch, and (2) combining this with proximity sensing. My main concern is that this seems like an evolutionary advance because it is already possible to achieve this same type of dual-mode response by combining known things in the literature. To be fair, the approach here is noted for its simplicity and sensitivity to touch, but at the same time, it does require external magnets (and if I understand, those magnets have to be magnetized in a particular polarity).

Aside from this major concern, the following are other thoughts:

1. Is it correct that the sensing requires a powerful external magnet?
2. I may have missed it, but I never saw quantification of the value of d_2 . In other words, how close in proximity does the sensor need to be to the magnet? This affects how close the sensor needs to be to achieve sensing.
3. Why is a wooden finger used instead of a real finger? Due to thermal sensitivity?
4. Is 'sensorics' a word?
5. Is there a reference for the "softest human touch" referred to on page 7?
6. From Figure 3c, it seems like the difference between pyramids+air versus just air gap isn't that different? (aside from the noted response time). Correct?
7. It might help to clarify the size of the pyramids and the GMR sensor in the main text. When you push down, how many pyramids touch the sensor (for example)?

Reviewer #3 (Remarks to the Author):

Dear Authors

To the best of my knowledge the contribution presented here is novel by a group that is at the front of their field (no pun here!). The use of magnetic field effects for pressure and touchless sensing in a single platform is a good idea that could see its integration into wearable technology. The demonstration of touch sensitivity with the flower example is quite compelling.

This work is worthy of publication. But I feel that the authors must report its possible drawbacks and what needs to be explored further before it would be fit for use. Here is a list of things which leave me feeling uneasy about the work:

1. The experiment depicted in Fig. S9 and its associated movie is an ideal case where the external magnet is near ideally aligned with respect to the axis of approach to the sensor device. What if there was a small angle of misalignment with respect to the axis of approach? Would this confuse the algorithm for touch? Would the results be much different? Has this been explored?
2. Does the external magnetic device need to be optimized for it to work well? Perhaps I missed something but there seems to be a paucity of experimental data. Can there be a less than optimal combination of magnet and sensor. Maybe this is not necessary - I welcome a comment from the authors about this.
3. How robust is the sensing system in the presence of stray fields or an extended field? I know that one of the selling points is the use of magnetic field - you can decide where to put the

magnet. But if we want to sense an extended object won't we need to have a longer magnetic zone or series of magnets arrayed along the surface? So I would have liked to see an experiment with a range of targets of different sizes and some perhaps near to each other or at least a good explanation of why this isn't necessary. Once again, perhaps I missed something.

Finally there are few typos and the grammar is good. A double-check would be good - Look at Fig. S5 for instance feild should be field.

A good piece of work with a little polishing to do.

Author's Reply

We would first like to thank the reviewers for their invaluable and constructive comments, all of which have been addressed with utmost care. The revised points are highlighted in blue in the manuscript. Our itemized responses to all the reviewer's comments are below.

Reviewer #1

In this paper, Ge et. al. present a bimodal electronic skin that can detect tactile and touchless interactions with an external magnet. The sensor composed of a GMR sensor and a soft composite magnet with micro pyramid arrays. The manuscript is nicely written, and the figures are beautiful. The idea of combining tactile and touchless interactions using all magnetic sensors is interesting, which could expand the sensing regime of conventional electronic skins. I also like the optimization of the pyramid shape soft magnet to optimize the sensitivity for the GMR sensor.

We thank the reviewer for this positive evaluation of our work. We revised the manuscript accordingly to her/his feedback, namely

- 1/ we carried out additional experiments to validate the angle sensing possibility of the m-MEMS platform
- 2/ we demonstrate that the pressure sensing functionality remains even if the sensor platform is immobilized on soft substrates like a human finger
- 3/ we elaborated on the sensing resolution in the in-plane and out-of-plane directions

However, the system has a few major challenges for the exciting future applications authors visioned in the paper.

1, Limited sensing pose in the touchless sensing mode.

In the demonstrations (figure 2), the magnetic fields generated by the magnetic film and magnetic object are always the opposite. This requires a fixed angle between the sensor and the external magnet. However, in the applications of surgical operations and robotic grasping (as mentioned in the introduction), the orientation sensing of the target object is crucial for manipulation. How can this system sense the orientation? For example, in figure 2, what will happen if the finger approaches the flower paddle from the back or from the side? Can the sensor still separate the tactile and touchless modes? Without the capability of sensing the orientation, the system simply just using a well-studied magnetic force sensor (embed magnetic in a soft matrix with a magnetic field sensor) to sense an external magnet.

We thank the reviewer for this crucial remark. Indeed, our compliant m-MEMS platform has the rotation sensing functionality and can discriminate the angle with respect to the direction of an external magnetic field.

Being motivated by the remark of the reviewer, we carried out a set of experiments to explore this functionality in the revised version of the manuscript. In brief, the sensing mechanism is based on the compensation of the magnetic field of the object and the

built-in magnetic field of the compliant magnetic patch integrated in the m-MEMS platform. In this respect, the operational principle of the angle sensor is similar to the proximity sensing mechanism as in touchless mode.

Figure R1a below illustrates the proximity sensing mechanism as discussed in the original manuscript. With decreasing distance, the built-in magnetic field is compensated by the magnetic field of the object. Similarly, the rotation of the sensor platform with respect to the external magnetic field leads to the compensation of the built-in magnetic field. In this respect, as illustrated in Figure R1b, the resistance of the compliant GMR sensor is determined by the absolute value of the total in-plane magnetic field. Taking the absolute magnetic field of the sensor and object being a and b , respectively, and θ - the rotation angle, the absolute value of the *total* magnetic field, $|H|_{\text{total}}$, can be expressed as follows:

$$|H|_{\text{total}} = \sqrt{a^2 + b^2 - 2ab \cos \theta}$$

With the increase ($0 \rightarrow 90^\circ$) or decrease ($0 \rightarrow -90^\circ$) of the rotation angle θ , the $|H|_{\text{total}}$ will increase. Therefore, the resistance of the sensor decreases with the increase of the rotation angle (absolute value).

Figure R1. (a) Schematic illustration of the compensation effect of the built-in magnetic field upon approaching to a magnetized object. (b) Schematic illustration of the compensation effect of the built-in magnetic field upon rotating the m-MEMS platform with respect to the stray field of a magnetized object. (c) A setup to study the angle sensing functionality of the m-MEMS platform. (d) The change of the electrical resistance of the compliant GMR sensor upon approaching to and rotating with

respect to a permanent magnet. R_0 denotes the initial resistance of the GMR sensors. $\Delta R = R_0 - R$, where R is the actual value of the electrical resistance when the sensor is exposed to an external magnetic field.

To demonstrate the rotation sensing functionality of the developed m-MEMS platform, we study experimentally the rotation dependent resistance change of our sensor using a setup shown in Figure R1c. The signal evolution upon touchless interaction is shown in Figure R1d. The resistance of our sensor increases when the sensor approaches the object from about 7 cm to 3 cm. Then, we hold the sensor at the position of 3 cm and rotate the sensor from 0 to 50°. This corresponds to a typical range of angular displacement of a human pointing finger. We can see that the resistance of our sensor decreases ($\Delta R/R_0$ increases) versus the increase of the rotation angle. In Figure R1d, we can also see that the rotation-induced readout signal is still within the range assigned by the algorithm as being touchless interaction. Therefore, the signal of the rotation-based touchless interaction is also separated from the signal of the tactile interaction. We note that discrimination of the angle sensing and proximity sensing in this demonstrator is done at the software level (angle sensing is performed at a constant distance to the object of interest). However, magnetic field sensorics allow for the unambiguous discrimination between proximity and angle sensing functions based on complementary GMR sensors (GMR multilayers combined with GMR-based spin valve sensors [Science Advances 4, eaao2623 (2018)]).

We add a discussion about the now explored rotation sensing functionality of the m-MEMS platform to the revised manuscript (Supplementary Fig. S19, Movie S5).

With respect to the question regarding the interaction with a petal (Figure 2 of the main text): if the finger approaches the flower petal from the back or from the front side, the sensing responses *might be* different. If the magnetic stray field is in-plane isotropic and varies with the distance to the object only, then the readout will be exactly the same. This is a usual example of rod-like permanent magnets or current-carrying wires. The *quantitative* difference in the signal will be only if the magnetic stray field profile is anisotropic in all three directions but the field lines will not change the direction with respect to the magnetic field of the built-in magnet. These quantitative differences are rather easy to correct for using the selection algorithm. Only when the direction of the external field will have a different sign to the one of the built-in magnetic patch, the *qualitative* differences are expected as discussed in the originally submitted manuscript.

Therefore, the proper choice of a magnetic object with isotropic in-plane magnetic stray fields is beneficial for our concept. Although it may be considered as a limitation, the isotropic in-plane stray fields are typical for any permanent magnet of elongated shape. This statement is always true even for the case of not very symmetric yet elongated magnetic objects when measured in a far field, where any object can be approximated as a dipole. The interaction we are exploring in our work happens in a far field. Therefore, the concept is rather generally applicable.

This discussion is added to the manuscript (page 7).

Furthermore, we note that this issue with a qualitative difference of the readout signal if the external field is not properly oriented with respect to the built-in field is *not* a fundamental limitation of the proposed concept of touchless interaction but just a consequence of the used magnetic sensor. For example, a Hall effect sensor (instead of GMR) would allow following the polarity of the magnetic field, and is subject of further studies. This will solve the issue with the possibility to discriminate the direction of the applied magnetic field.

In this respect, we note that for some future applications, we are focusing on developing a platform sensitive to all three components of an external magnetic field.

This discussion is added to the manuscript in the outlook section (page 21).

2, Number of magnetic objects this sensor can distinguish.

In programmable interactions (mentioned in the abstract), the electronic skin needs to interact with multiple target objects. I wonder how the current e-skin distinguish different magnetic objects. Is that possible to separate magnetic objects by strength, orientation, or combinations of multiple magnets?

With “programmable interaction” we here mean “signal-programmable interaction” that allows us to identify objects that we need to interact with. Furthermore, we can determine the sensing response of our sensor to the objects by adjusting the magnetic field of our sensor and the magnetic field of the objects. We apply permanent magnetic field patches to a petal and a glass plate (both objects are non-magnetic) to specify them as the objects of interest for the interaction using our m-MEMS platform. Furthermore, by adjusting the magnetic field of our sensor and the objects, we enabled our sensor with two different sensing response to the objects: (1) Tactile-prior interaction meaning low proximity sensitivity but higher pressure sensitivity (Figure 2 of the main text); (2) Touchless-prior interaction meaning high proximity sensitivity but low pressure sensitivity (Figure 5 of the main text).

In this sense of “programmable interaction” our magnetism-enabled interactivity has advantages over capacitive bi-modal sensors, which can hardly specify the object of interest and tailor the sensing behavior for desired touchless interaction. With these advantages, our sensor is able to separate the signal of two different interaction modes in real time. To our knowledge, this is the first sensor than can distinguish two interaction modes just based on one source of electric signal (it refer to the change of resistance of the GMR sensor in this work).

Although initially not considered, we were inspired to address the comment of the reviewer on the possibility to recognize multiple magnetic objects: if magnetic objects are magnetically different (different field strength, different symmetry of the magnetic field, different field gradients) and they are positioned relatively far away from each other, then our sensing platform can be trained to recognize these objects. This is the case demonstrated in figure 2 and figure 5. In the first case when a thin magnetic patch is used, the field is weak and interaction starts at close distance. In the

second case (standard permanent magnet), the field is rather strong and interaction starts at larger distance to the object. The spatio-temporal variation of the signal will be different when approaching these two objects. These differences can be used as unique fingerprints of each magnetic objects. These spatio-temporal maps can be constructed for different magnetic objects and used for their identification.

More challenging is the situation when different magnetic objects are located close to each other such that the magnetic field sensor will detect superposition of both stray fields. In this case, the reconstruction is possible as well. However, in addition to the measurement of magnetic field strength, the sensor should be able to analyze gradients of fields. In the simplest configuration, two-sensor-arrays could be used to assess the spatial evolution of the field gradients. This method further benefits from the displacement of the magnet with respect to sensor array. The known example of such reconstruction is magnetic resonance imaging or tomography. Although definitely feasible to be realized, these algorithms and realizations are rather involved.

Still, to avoid complication with the reconstruction, we appeal to the fast decay of the magnetic field strength of standard permanent magnets. For typically used small pieces, the relevant field of about 1 mT would extend to the distance of about 1-2 cm. Practically, it means that if the magnets are separated by the distance of several cm only, they will not strongly interact, which will make it possible to treat these objects as being independent from the position of view of our wearable m-MEMS platform. In this case, in typical settings of the examples given in our manuscript, the two objects can be uniquely recognized using “trained” magnetic field sensor as described above.

This discussion is added to the manuscript in the outlook section (page 21).

In the manipulation of virtual objects, as shown in Figure 5a, the finger can select virtual option menu in the touchless mode by swipe the finger. I assume the sensor chooses the option by the magnetic field strength. If that is true, moving away the sensor will have the same effect, making it unfit to manipulate virtual objects due to limited spatial resolution. The author can provide a systematic discussion on the spatial resolution (position and orientation) of this sensor in the touchless mode, and comment on how to solve this issue.

The sensor chooses the option by evaluating the magnetic field strength. Furthermore, it is correct that moving the sensor away from the source of the magnetic field will result in the decay of the field, which will be reflected in the change of the sensor resistance. If the magnetic stray field is isotropic (3D isotropic magnet is used like a sphere), then shifting the magnet in vertical (z direction) and horizontal (x or y direction) would result in the same signal change. Only in this case, it will not be possible to distinguish between z-movement and lateral displacement.

While this example is not impossible, it is rather special. It is more common to use standard permanent magnets, which have spatial anisotropy of magnetic stray fields. In this case, a displacement of the sensor in different planes will have different strength of the signal change. Majority of platelet-shaped magnets have rather homogeneous field in their plane (not axially symmetric but homogeneous in the plane

of the patch) and the field decays fast in the perpendicular direction. Those magnets are beneficial for the realization of the concept shown in figure 5. In this respect, the manipulations shown in figure 5 can be uniquely interpreted.

Regarding the spatial resolution: the sensors used in this work are macroscopic. Their lateral dimensions are chosen to be in the range of the typical area of a fingertip. This is done in purpose to assure averaging of spatial changes of the external magnetic fields on the distances relevant for the finger motion. Those displacements are of about the size of the finger.

Still, it is definitely possible to miniaturize the sensors. However, the main idea for this miniaturization would be related to the possibility to realize a sensor array. The array will consume the same area of a fingertip but will contain more sensor elements. The combination of these sensors will allow to track the change of the magnetic field strength in space and in this way will provide access to magnetic field gradients. Realization of gradiometers would be another important step towards even higher fidelity of the interpretation of the magnetic field changes upon touchless and tactile manipulation.

3, Sensing on a soft substrate and spatial of the sensors

Considering the size of the sensor (~ 5mm), the surface under the sensor region cannot be simply considered rigid. In the demonstrations (figure2, 4, 5), the sensors are wrapped around a rigid wooden finger. I am just wondering what will happen on a human finger, and how much the soft substrate can impact tactile sensing signals.

We thank the reviewer for this insightful remark. Indeed, the hardness of the underlying substrate influences the quantitative readout of our sensor. Qualitatively, everything remains the same and the sensor works as a pressure sensor. In this respect, our sensor responds similarly to other flexible pressure sensors. Therefore, the typical approach to achieve *quantitative* comparison of different flexible sensors is to perform characterization on rigid substrates. Up to now, there is no standard criteria for the measurement of flexible pressure sensor on soft substrates. Therefore, quantitative characterization of our pressure sensor is done on rigid support. In this case, the data we provide can be compared to the literature reports.

In our particular case: when applied to a soft substrate, like a fingertip, the shape deformation of the m-MEMS upon pressing will change the magnetic field at the position of the GMR sensor. This deformation will be different compared to the case when the sensor is applied to a rigid substrate like wooden finger. Still, conceptually/qualitatively nothing will change and the m-MEMS platform will act as a pressure sensor. This statement is supported by the new experimental data shown in Figure R2 below. Here, we immobilized our m-MEMS platform on a finger (Figure R2a). The resistance change of our sensor on the finger skin could distinguish normal touch and light touch (Figure R2b and Move R1).

This information is added to the revised manuscript as Supplementary Fig. S22 and Movie S6.

Figure R2. Demonstration of the pressure sensing functionality of the developed m-MEMS platform fixed on a finger skin. R_0 denotes the initial resistance of the m-MEMS platform. $\Delta R = R_0 - R$, where R is the real time resistance.

The spatial sensing resolution at the x-y plane (parallel to the skin) is significantly smaller than the z direction (perpendicular to the skin). Can miniaturization solve this issue? ((multi-sensor resolution is different with the touchless moving resolution of finger)) Maybe the authors can comment on this as a perspective.

We thank the reviewer for this insightful comment. Indeed, the use of a single macroscopic GMR sensor limits the in-plane spatial resolution. As mentioned by the reviewer, this issue can be solved in two ways. Either miniaturizing the sensor or/and going for a sensor matrix. Both solutions are established for rigid counterparts of magnetic field sensors. At the same time, the aspect of miniaturization of GMR sensors on compliant foils is yet to be addressed in the community.

We implemented this discussion on page 15.

Here are also some minor comments:

1, Some expressions are quite dramatic, e.g. “Practical electronic embodiments would benefit most ...” “... our m-MEMS platform is a key milestone ...”

2, I don’t understand “magnetic functionality”, I suggest changing it to “embedded magnet”.

We revised the manuscript accordingly to avoid any possible misinterpretation.

Reviewer #2:

This manuscript describes the use of a GMR (giant magneto resistive) sensor, which changes resistance in response to magnetic fields. Here, the magnetic fields come from two sources. One source is an external magnet, which can be detected in proximity without direct contact. The second is from an array of magnetic (soft) pyramids that hover over the sensor. When this array is pressed, the pyramids are brought into closer proximity to the sensor and can therefore be used for sensing touch. This type of bi-modal sensing is of interest for prosthetics and human-machine interactions, for example.

We thank the reviewer for her/his insightful comments, which were used to refine the manuscript. In brief:

- 1/ we extended the discussion on the strength of the external magnetic field, which is needed to operate the m-MEMS platform
- 2/ we provide information on the measurement of the pressure when the m-MEMS platform is mounted on a human finger
- 3/ we elaborate on the determination of the distance d_2 and provide practical considerations on the key parameters, which define this distance

Although the paper is well-written with nice figures and demonstrations, my concern is the following. The use of GMR for sensing remote magnetic fields has been done previously. The use of pyramids for sensing touch has also been done previously, but using capacitance. If I understand, the novelty is (1) the use of pyramids for GMR sensing touch, and (2) combining this with proximity sensing. My main concern is that this seems like an evolutionary advance because it is already possible to achieve this same type of dual-mode response by combining known things in the literature. To be fair, the approach here is noted for its simplicity and sensitivity to touch, but at the same time, it does require external magnets (and if I understand, those magnets have to be magnetized in a particular polarity).

We agree with the reviewer that GMR sensors and pyramid-based touch sensors have been reported. Relevant references on different technologies are provided in the originally submitted manuscript.

As the reviewer noted, the novelty of our work is *not* in the individual components but rather in the new concept of interaction with objects. We put forth a concept of combined touchless and tactile interactions based on the use of magnetic fields. To validate our concept, we designed and fabricated a novel functional device – compliant magnetic MEMS, which was not reported in literature. With this device, we demonstrate how the electrical readout signal of the GMR sensor can be treated to achieve unambiguous separation of touchless and tactile interaction *with a single sensor*.

In this respect, both the realization of the m-MEMS and the algorithm of the data analysis are key conceptual aspects of our work.

Without this algorithm, a simple combination of a pyramid-structured pressure sensor and a magnetic sensor together will, most likely, fail to unambiguously discriminate different interaction modes. This is one of the issues faced by previously reported multi-mode sensors. The electric signals of different sensing modes overlap with each other, making it challenging for a single sensor unit to discriminate different sensing modes by relying on the same source of electric signal.

With our concept, we successfully solved this long-standing problem. Our m-MEMS together with a proper algorithm of data treatment can unambiguously discriminate signals of different interaction modes into different regions and realize the discrimination of two sensing modes in real time, while being based on the same source of electric signal (only the resistance change of the GMR sensor). The realization of the entire concept is the key achievement of our work.

Indeed, the interaction mode is based on the use of magnetic stray fields. Therefore, to enable touchless interaction with external objects, those objects should be magnetic. Tactile interaction can be done also with non-magnetic objects as the m-MEMS platform contains a magnetic field source realized as embedded magnetic patch. The use of magnetic objects provides an important advantage of having the possibility to address objects selectively. Typically, objects surrounding us are non-magnetic. In this case, by adding magnetic decoration to the wall of i.e. a room (say, permanent magnets in a wall), we will not disturb the everyday arrangement of objects but will gain an important feature of having an additional interaction channel via magnetic fields. This additional information channel allows simplifying interaction with objects. For instance, in augmented reality settings, the use of m-MEMS allows to reduce the number of physical “clicks” to manipulate properties of virtual devices.

Aside from this major concern, the following are other thoughts:

1. Is it correct that the sensing requires a powerful external magnet?

We appreciate this important remark of the reviewer. To make the concept of m-MEMS viable, magnetic field sources should be as weak as possible to avoid any harmful effect on humans and on electronics. At the same time, they should be substantially stronger than the geomagnetic field of about 50 μT . The latter is important to avoid undesirable interferences with small magnetic fields from buildings, electrical power lines etc.

Considering that a typical field of a small piece of a permanent magnet is in the range of 1 mT at the distance of several cm, we decided to have our platform working in this field range.

Therefore, small pieces of commercially available permanent magnets (like those used as fridge magnets) can be used to enable the magnetic field-based interaction with the m-MEMS platform. No powerful external magnet is necessary for our concept.

2. I may have missed it, but I never saw quantification of the value of d_2 . In other words, how close in proximity does the sensor need to be to the magnet? This affects how close the sensor needs to be to achieve sensing.

We initially did not quantify the value of d_2 because this value can be tuned in a wide range depending on the use case. For example, d_2 is about 5 mm for the case of the demonstrator with a flower petal (Figure 2 of the main text), and is about 30 mm in the demonstrator shown in Figure 5 of the main text. The value of d_2 is determined by the gradient of the magnetic field of the object. As shown in the Figure R3 below, the magnetic field at the plane of the object's surface (H_o) should be smaller than the magnetic field at the plane of the GMR sensor (H_s). d_2 is the distance between the object surface and the plane where the magnetic field is lowered to the sensing limit of the GMR sensor (named as H_{sl} , which is in our specific case of about 0.05 mT). Therefore, d_2 increases with the decrease of the gradient between H_o and H_{sl} .

The gradient of the magnetic field can be tuned by choosing a proper design of a magnet or using arrays of magnets to shape the stray field at will (in the spirit of the remark of the third reviewer).

To address this aspect, we added a discussion in the revised manuscript (Fig. S18).

Figure R3. Schematic illustration of the key factor that determines the value of d_2 .

3. Why is a wooden finger used instead of a real finger? Due to thermal sensitivity?

When placing the m-MEMS device consisting of only one GMR sensor on the finger, there are thermalization effects as shown in figure R2 above (gradual decay of the baseline with time). However, these are readily tackled using standard approaches relying on sensor bridges. For this sake, in figure 5 of the main text, we used the Wheatstone bridge sensor arrangement to eliminate the temperature effects [Science Advances 4, eaao2623 (2018)].

The primary idea of using a wooden finger was to have a rigid substrate. Like this, the measured pressure data is quantitatively comparable to the literature results. In this regard, the hardness of the underlying substrate influences the quantitative readout for all flexible pressure sensors. Therefore, to avoid complications with the data interpretation, we here used a wooden model.

However, as we show now in figure R2 above, the concept works well even if the m-MEMS platform is positioned on a real human finger.

This information is provided in the revised manuscript as Supplementary Fig. S22 and Movie S6.

4. Is 'sensorics' a word?

This term seems frequently used in the community in papers but also in many books, e.g. "Fundamentals of Piezoelectric Sensorics" and "Piezoelectric Sensorics".

5. Is there a reference for the "softest human touch" referred to on page 7?

Yes, this term was mentioned in [*Nat. Mater.* 2010, 9, 790]. This paper is added to the reference list of the revised manuscript. We thank the reviewer for noticing this.

6. From Figure 3c, it seems like the difference between pyramids+air versus just air gap isn't that different? (Aside from the noted response time). Correct?

Indeed, the difference between pyramids + air gap and just air gap is not so large regarding the pressure sensitivity. In addition to the aspect of pressure sensitivity, the key contribution of pyramids is their anti-sticking effect. This has a major impact on the speed of the sensor. With the pyramids, the response time of our sensor is as fast as 75 ms (Figure 3e in the main text). In contrast, the sensor without pyramids cannot return to the original value of resistance even within 60 s after the release of pressure (Fig. S13).

7. It might help to clarify the size of the pyramids and the GMR sensor in the main text. When you push down, how many pyramids touch the sensor (for example)?

We thank the reviewer for her/his suggestion. Following this remark, we provide the dimensions of the pyramids and GMR sensor in the revised manuscript (Figure S2). The base plane of the pyramid is 50 μm by 50 μm . The height of the pyramid is 35 μm . The distance between adjacent pyramids is 100 μm . The number of pyramids in the magnetic patch is about 1255. From the area of the GMR sensor and taking into account the homogeneous coverage of the magnet with pyramids, we estimate that about 270 pyramids are in contact with the GMR sensor.

This information is provided in the caption of Figure S2.

Reviewer #3:

Dear Authors

To the best of my knowledge the contribution presented here is novel by a group that is at the front of their field (no pun here!). The use of magnetic field effects for pressure and touchless sensing in a single platform is a good idea that could see its integration into wearable technology. The demonstration of touch sensitivity with the flower example is quite compelling. This work is worthy of publication.

We appreciate the positive evaluation of our work by the reviewer. Following her/his remarks, we modified the manuscript as follows:

- 1/ we extended the outlook section and especially elaborate on possible drawbacks of the technology in its present realization and suggested possibilities to overcome the limitations
- 2/ we elaborated on the magnetic field of the external object, which is needed to assure discrimination of the operation modes using the m-MEMS platform
- 3/ we extended discussion on the envisioned use case scenario.

But I feel that the authors must report its possible drawbacks and what needs to be explored further before it would be fit for use. Here is a list of things which leave me feeling uneasy about the work:

Following the suggestion of the reviewer, we elaborated on possible drawbacks and on possible routes to address them.

The discussion is in the outlook section of the revised manuscript.

In brief, we identify several important challenges:

- 1/generic magnetic stray fields are 3D: therefore, to address this aspect, the developed here m-MEMS platform would need to be extended to be able to measure all 3 components of magnetic field. This would necessarily require integration of other sensors into the m-MEMS package including those based on Hall effect.
- 2/ Addressing the possibility to interact with several magnetic objects: this task would require the development of a new logic in the analysis software which will take into account not only spatial but also temporal variation of the readout signal.

1. The experiment depicted in Fig. S9 and its associated movie is an ideal case where the external magnet is near ideally aligned with respect to the axis of approach to the sensor device. What if there was a small angle of misalignment with respect to the axis of approach? Would this confuse the algorithm for touch? Would the results be much different? Has this been explored?

We thank the reviewer for this insightful remark. The idealized case is shown in Fig. S9, used to assure *quantitative* analysis of the sensor performance in different modes.

However, all showcase demonstrators are done using the developed m-MEMS attached to a wooden finger or a human finger. In these settings, the ideal positioning of the m-MEMS is not possible and misalignments are unavoidable. Still, the sensor performs well and discrimination of the sensing modes can be done based on the proposed algorithm. This is especially evident in the discussion around figure 5 of the main text and corresponding video. We repeated the approaching and sweeping processes three times. As seen in the video, we did not strive to keep the wooden hand stable but allow shaking. Independent of these dynamic misalignments, the platform works reliably in both operation modes.

Therefore, realistic misalignments, which appear upon standard manipulations, do not result in the failure of the discrimination of the operation mode. We note that in the case of the example shown in figure 5 these misalignments are sizable in the range of about 5-10°.

2. Does the external magnetic device need to be optimized for it to work well? Perhaps I missed something but there seems to be a paucity of experimental data. Can there be a less than optimal combination of magnet and sensor. Maybe this is not necessary - I welcome a comment from the authors about this.

We indeed ideally require the external magnets to be “fitting” in terms of the strength, spatial profile and the direction of their magnetic field. Firstly, the direction of the field of the external magnet should be opposite to that of the built-in magnetic field of the m-MEMS platform. Then, the magnetic field strength surrounding the object of interest should be adjusted to the proper value for different interaction tasks, i.e. small magnetic field (Fig. S6) for tactile-prior interaction and strong magnetic field (Fig. S16) for touchless-prior interaction. Furthermore, the spatial profile of the magnetic field should be properly chosen to assure the possibility to uniquely interpret displacements of the m-MEMS platform (say, homogeneous field in-plane and decay in the out-of-plane direction).

Although seemingly limiting, the concept remains rather general as those magnetic fields are typically obtained using standard permanent magnets. Say, magnetic field patches provide homogenous in-plane field (in a certain area) and rapidly decaying field in z-direction. If axial symmetry of the magnetic field is needed, it is more appropriate to use elongated magnets, which in the far field will provide the needed configuration of magnetic fields. We emphasize that these are standard magnets readily evaluable commercially. Therefore, the practical implementation of the concept does not require any special development of permanent magnets.

We note that the main goal of optimizing the combination of magnet and sensor is to separate the electric signal of tactile and touchless interaction modes. Without the optimization, the sensor also has proximity and pressure sensing functions, but it might be more challenging to distinguish the two interaction modes in real time. As is shown in the Figure R5 below (also see Movie R2), the signal of touchless and tactile interaction overlap. In this case, the sensor cannot distinguish the interaction mode in real time without knowing the history of the evolution of electric signal.

Figure R5. An example of the interaction performed under non-optimized magnetic field. R_0 denotes the initial resistance of the MEMS platform. $\Delta R = R_0 - R$, where R is the real time resistance.

3. How robust is the sensing system in the presence of stray fields or an extended field? I know that one of the selling points is the use of magnetic field - you can decide where to put the magnet. But if we want to sense an extended object won't we need to have a longer magnetic zone or series of magnets arrayed along the surface? So I would have liked to see an experiment with a range of targets of different sizes and some perhaps near to each other or at least a good explanation of why this isn't necessary. Once again, perhaps I missed something.

We note that the interaction concept is based on the interaction of magnetic field sensors with stray fields of magnetic objects. In this respect, it is essential to have the system robust with respect stray magnetic fields. As we work with the fields of standard permanent magnet pieces, our target magnetic field is in the range of 1 mT. Therefore, the sensor is also optimized to be sensitive in this field range. In terms of robustness of the sensor response, it is to assure that the external magnetic field does not overcome the saturation field of the magnetic field sensors. For instance, based on the sensor characterization (figure 1m of the main text), the sensor will be saturated in the field of, say, about 5 mT. Admittedly, this is rather strong magnetic field, which is not easy to achieve using small magnet pieces at relevant interaction distances.

Still, in line with the remark of the reviewer, when the external stray field will exceed 5 mT, the sensor platform will lose its sensitivity. The way around this issue is twofold:

1/ one can use a different magnetic sensor. Other GMR sensor can be used with a larger operation field range [Melzer et al., RSC Advances 2, 2284 (2012)], say Py/Cu multilayers coupled at the 1st antiferromagnetic maximum (in contrast to the Py/Cu

multilayers coupled at the 2nd antiferromagnetic maximum as used in this work). Alternatively, one can use a different sensor concept, say based on Hall effect. Those sensors are linear sensor of magnetic field and hence they do not exhibit the saturation behavior.

2/ one can use a magnetic patch built-in in the m-MEMS platform, which will exhibit a stronger magnetic field. In this case, the working point of the sensor can be shifted for several mT, which will effectively enhance the saturation field of the sensor with respect to the external magnetic field.

Following the remark of the reviewer regarding the use of arrays of magnets or elongated magnets: using elongated objects is of advantage as they provide favorable stray field provide in the far field. Using flat elongated objects (like stripes), one can achieve homogeneous fields in the plane of the magnet and decaying fields in the z-direction perpendicular to the stripe plane. This is of advantage to realize the concept shown in figure 5.

Magnetic objects are needed to enable the interaction in our system. We envision that they are placed such that they do not to disturb the optics of the room where they are placed.

Regarding using arrays of magnets: It might be of advantage to use array of magnets with the aim to shape the profile of the stray field. For instance, combining two bar magnets, it is possible to get not a dipolar but quadrupolar magnetic field profile. More complex field profiles might be of advantage to have them as fingerprints of a specific “magnetic knob” (identification of the magnetic object based on specific field profile). In any case, even when using an array of magnets, we still rely on the interaction with the resulting field of all of them. The reason is like this: magnetic fields of several magnetic objects will sum up (vector summation) and an individual magnetic field sensor can measure only the total magnetic field at the sensor location. In this respect, to discriminate between different magnets positioned next to each other (in the spirit of the question posed by the first reviewer), one would need to use magnetic gradiometers in combination with magnetic field sensors. Although technically realizable, these settings go beyond the current concept of interaction of the wearable m-MEMS with a point-like magnetic field source stemming from an external magnetic object.

To better transport the vision behind our concept, we added an additional schematics showing a use case (figure R6 below). The magnetic object of interest works like a virtual button, which enables interaction with a physical object, e.g. light, humidifier or similar device. The magnetic knob is located conveniently for its manipulation. When our m-MEMS platform approaches and presses the magnetic object, the electric signals will be analyzed by the wearable computer and be transformed to commands (i.e. select the device of interest, change the value). The commands will be send wirelessly to the device of choice (e.g. air conditioner) to change the actual setting of the device, say the temperature in the room. Therefore, we only need one magnetic object to realize manipulation with multiple devices.

We added this discussion to the outlook section and as a new supplementary Fig. S15.

Figure R6. Application concept of the m-MEMS platform.

Finally there are few typos and the grammar is good. A double-check would be good - Look at Fig. S5 for instance feild should be field.

The manuscript is revised to correct typos.

A good piece of work with a little polishing to do.

We thank the reviewer for these encouraging remarks. We revised the manuscript accordingly to her/his suggestions. We hope that the revised manuscript now matches the high standards of Nature Communications.

REVIEWERS' COMMENTS:

Reviewer #1 (Remarks to the Author):

I am satisfied with the changes to the paper. The basic idea is clearly explained, the feasibility is well-tested, and the application scenario makes sense.

Reviewer #2 (Remarks to the Author):

I thank the authors for their efforts to revise the manuscript and address concerns of the reviewers. It is very nice work and interesting that both proximity and touch can be sensed.

I don't have any lingering technical concerns, although I will be interested if Reviewer 1 is satisfied with the response since he / she raises a good point: in essence, these sensors measure one thing – resistance versus time, but many things can cause resistance to change and thus may not be possible to distinguish those effects in realistic settings.

My only overall concern is whether this really reaches the threshold of novelty needed for Nature Communications. The authors did a nice job of addressing the novelty in their response by pointing out the work incorporates two types of sensing (touch and proximity) in the same sensor.

That said, I think that using two "off-the-shelf" sensors (stacked on top of each other or placed adjacent) could do the same thing: For example, (1) a GMR could measure proximity to a magnet (via changes in resistance) and (2) a pyramid sensor could measure touch (via changes in capacitance), in a similarly sized device with similar overall mechanics. Admittedly, that would require two different sensors rather than one, so really the question is whether having these features in one device (versus two) is important. Having one sensor (versus two sensors) adds processing complexity to decipher multiple inputs from a single sensor. In addition, the sensor itself is rarely what limits the size of a device – instead it is the attached electronics needed to power, process, and communicate the data.

So the lingering question in my mind is not whether this is good work (it is great, in fact), but whether it really solves a problem.

Reviewer #3 (Remarks to the Author):

I am satisfied with the authors' response to my critique. I am happy to see this work published.

Reponse to the remarks of Referee 2

I thank the authors for their efforts to revise the manuscript and address concerns of the reviewers. It is very nice work and interesting that both proximity and touch can be sensed.

We thank the Referee for his/her positive feedback on our manuscript.

I don't have any lingering technical concerns, although I will be interested if Reviewer 1 is satisfied with the response since he / she raises a good point: in essence, these sensors measure one thing – resistance versus time, but many things can cause resistance to change and thus may not be possible to distinguish those effects in realistic settings. My only overall concern is whether this really reaches the threshold of novelty needed for Nature Communications. The authors did a nice job of addressing the novelty in their response by pointing out the work incorporates two types of sensing (touch and proximity) in the same sensor.

With respect to the first comment, we note that the first referee was satisfied with the revised manuscript and did not raise any further question. The referee 2 is right, we measure with our sensor resistance change with time. Therefore, the major novelty of this work is that it is possible to find conditions, where the resistance changes is qualitatively different for the touchless and tactile interaction modes. The key achievement is the demonstration that introducing the concept of “safety window”, we can clearly relate the positive resistance change with the tactile interaction and negative resistance change with the touchless interaction.

In this respect, the discrimination of the interaction modes is unambiguous.

That said, I think that using two "off-the-shelf" sensors (stacked on top of each other or placed adjacent) could do the same thing: For example, (1) a GMR could measure proximity to a magnet (via changes in resistance) and (2) a pyramid sensor could measure touch (via changes in capacitance), in a similarly sized device with similar overall mechanics. Admittedly, that would require two different sensors rather than one, so really the question is whether having these features in one device (versus two) is important. Having one sensor (versus two sensors) adds processing complexity to decipher multiple inputs from a single sensor. In addition, the sensor itself is rarely what limits the size of a device – instead it is the attached electronics needed to power, process, and communicate the data. So the lingering question in my mind is not whether this is good work (it is great, in fact), but whether it really solves a problem.

We agree with the concern of the referee that having one multifunctional sensor (instead of two sensors with a specific functionality) adds processing complexity to separate multiple outputs from a single sensor. Typically, the problem is related to the overlap of the readout signals from different stimuli. To accomplish the discrimination task, one possible way is to increase number of outputs from the single sensor unit (see, e.g., *Adv. Mater.* 2016, 28, 10459). An obvious drawback of this strategy is a more complex electronics for the analysis of the readout signals.

To solve this long standing problem, here, we put forth a novel strategy allowing to separate unambiguously the sensor readout when actuated by different stimuli (touchless and tactile interaction modes) in real-time without using additional outputs. The analysis of the electric signal is realized in the same way as for a standard single sensor (single magnetic sensor or pressure sensor). No complex electronics is needed. The main finding of the manuscript is that

the use of magnetic MEMS enables a unique feature related to a *qualitative* difference in the signal level when operating in touchless (signal change is negative) or tactile (signal change is positive) modes.

The motivation to use one multifunctional sensor unit compared to an assembly of many dedicated sensors is as follows: Instead of integrating multiple single-function sensors, the combination of multiple functions in a single sensor greatly reduces the fabrication and integration complexity and thus decrease the cost of the electronic skins to meet the requirement of wide range of applications. Furthermore, it enables the intriguing direct and ultrafast communication between different functions without external electronics and signal processing.